# SIMPLICIAL COMPLEX NETWORKS

## ABSTRACT

Universal approximation property of neural networks is one of the motivations to use these models in various real-world problems. However, this property is not the only characteristic that makes neural networks unique as there is a wide range of other approaches with similar property. Another characteristic which makes these models interesting is that they can be trained with the backpropagation algorithm which allows an efficient gradient computation and gives these universal approximators the ability to efficiently learn complex manifolds from a large amount of data in different domains. Despite their abundant use in practice, neural networks are still not well understood and a broad range of ongoing research is to study the interpretability of neural networks. On the other hand, topological data analysis (TDA) relies on strong theoretical framework of (algebraic) topology along with other mathematical tools for analyzing possibly complex datasets. In this work, we leverage a universal approximation theorem originating from algebraic topology to build a connection between TDA and common neural network training framework. We introduce the notion of *automatic subdivisioning* and devise a particular type of neural networks for regression tasks: Simplicial Complex Networks (SCNs). SCN's architecture is defined with a set of bias functions along with a particular policy during the forward pass which alternates the common architecture search framework in neural networks. We believe the view of SCNs can be used as a step towards building interpretable deep learning models. Finally, we verify its performance on a set of regression problems.

## 1 INTRODUCTION

It is well-known that under mild assumptions on the activation function, a neural network with one hidden layer and a finite number of neurons can approximate continuous functions. This characteristic of neural networks is generally referred to as the universal approximation property. There are various theoretical universal approximators. For example, a result of the Stone-Weierstrass theorem Stone (1948); Cotter (1990) is that multivariate polynomials are dense in the space of continuous real valued functions defined over a hypercube. Another example is that the reproducing kernel Hilbert space (RKHS) associated with kernel functions with particular properties can be dense in the same space of functions. Kernel functions with this property are called *universal kernels* Micchelli et al. (2006). A subsequent result of this theory is that the set of functions generated by a Gaussian process regression with an appropriate kernel can approximate any continuous function over a hypercube with arbitrary precision. Although multivariate polynomials and Gaussian processes also have this approximation property, each has practical limitations that cause neural networks to be used more often in practice compared to these approaches. For instance, polynomial interpolations may result a model that overfits to the data and suffers from a poor generalization, and Gaussian processes often become computationally intractable for a large number of training data Bernardo et al..

Neural networks, with an efficient structure for gradient computation using backpropagation, can be trained using gradient based optimization for large datasets in a tractable time. Moreover, in contrast to existing polynomial interpolations, neural networks generalize well in practice. Theoretical and empirical understanding of the generalization power of neural networks is an ongoing research Novak et al. (2018); Neyshabur et al. (2017).

Topological Data Analysis (TDA), a geometric approach for data analysis, is a growing field which provides statistical and algorithmic methods to analyze the topological structures of data often referred to as point clouds. TDA methods mainly relied on deterministic methods until recently where

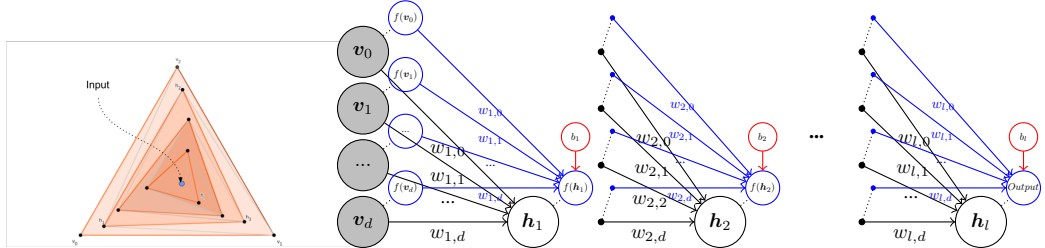

Figure 1: General view of an SCN: *Left*: without applying any transformation to its input, an SCN locates the position of input in the input space through a set of nested simplexes. *Right*: architecture of an SCN: $\boldsymbol{v_i}$ are a given set of visible vertices of a primary simplex in the input space that the sample falls inside. A network of hidden vectors $\boldsymbol{h_i}$ are then used to parameterize a sequence of nested simplexes for locating the input. Each $\boldsymbol{h_i}$ is a convex combination of a subset of its preceding vectors. In parallel, another network is used to generate SCN's output utilizing the output of SCN at all $\boldsymbol{v_i}$ and $\boldsymbol{h_i}$, combined with a group of bias values $b_i$.

statistical approaches were proposed for this purpose Carriere et al. (2017); Chazal & Michel (2017). In general, TDA methods assume a point cloud in a metric space with an inducing distance (e.g. Euclidean, Hausdorff, or Wasserstein distance) between samples and build a topological structure upon point clouds. The topological structure is then used to extract geometric information from data Chazal & Michel (2017). These models are not trained with gradient based approaches and they are generally limited to predetermined algorithms whose application to high dimensional spaces may be challenging Chazal (2016).

In this work, by leveraging geometrical perspective of TDA, we provide a class of restricted neural networks that preserve the universal approximation property and can be trained using a forward pass and the backpropagation algorithm. Motivated by the approximation theorem used to develop our method, Simplicial Complex Network (SCN) is chosen to refer these models. SCNs do not require an activation function and architecture search in the way that conventional neural networks do. Their hidden units are conceptually well defined, in contrast to feed-forward neural networks for which the role of a hidden unit is yet an ongoing problem. SCNs are discussed in more details in later sections.

Our contribution can be summarized in building a novel class of neural networks which we believe can be used in the future for developing deep models that are interpretable, and robust to perturbations. The rest of this paper is organized as follows: Section 2 is specified for the explanation of SCNs and their training procedure. In section 3, related works are explained. Sections 4, 5, and 6 are specified to experiments, limitations, and conclusion.

## 2    SIMPLICIAL COMPLEX NETWORKS

We first describe some necessary notation in the section 2.1. In section 2.2, we discuss the barycentric subdivision and the simplicial approximation theorem. In section 2.3, we modify the barycentric subdivision in order to develop an approach that allows us to learn a subdivision of the input space into small simplexes. We then introduce a general framework for defining an SCN for regression tasks that simultaneously learns a subdivision of the input space and a piece-wise linear mapping where the linear pieces are defined on each simplex of the subdivision.

### 2.1    NOTATION

We consider a dataset $D = \{(\boldsymbol{x}^{(i)}, \boldsymbol{y}^{(i)})\}_{i=1}^N$ of $N$ i.i.d. input/output pairs, where for each $1 \leq i \leq N$, $\boldsymbol{x}^{(i)} \in \mathbb{R}^d$ and $\boldsymbol{y}^{(i)} \in \mathbb{R}^k$. Let $f_\theta$ denotes a mapping from the input to the output space in which $\theta$ represents its parameters. In a regression task, we wish to minimize $J(\theta) = \mathbb{E}_{(\boldsymbol{x}^{(i)}, \boldsymbol{y}^{(i)}) \sim D}[\mathcal{L}(f_\theta(\boldsymbol{x}^{(i)}), \boldsymbol{y}^{(i)})]$, where $\mathcal{L}$ is a given loss function.

We indicate a $d$ dimensional simplex ($d$-simplex) with vertices $\boldsymbol{v}_0, ..., \boldsymbol{v}_d$ with the notation $[\boldsymbol{v}_0, \boldsymbol{v}_1, ..., \boldsymbol{v}_d]$. For simplicity of the presentation, we further assume that each $\boldsymbol{x}^{(i)}$ lies inside

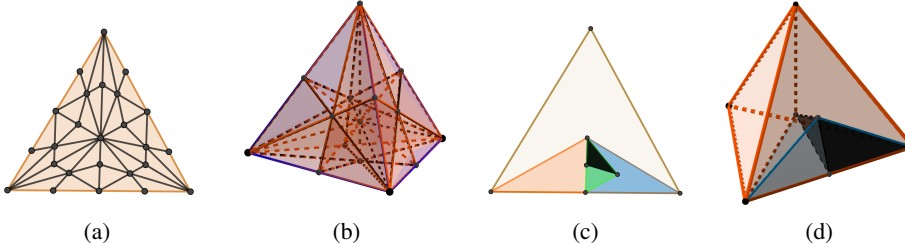

Figure 2: (a) Second barycentric subdivision of a 2-simplex. (b) First barycentric subdivision of a 3-simplex. (c), (d) Nested set of simplexes to get a sample simplex (shown in black) from: (c) the second barycentric subdivision of a 2-simplex, (d) the first barycentric subdivision of a 3-simplex.

a $d$-simplex $\sigma = [\boldsymbol{v}_0, \boldsymbol{v}_1, ..., \boldsymbol{v}_d]$, where $\boldsymbol{v_0}$ is the origin, and $\boldsymbol{v}_1, ...\boldsymbol{v}_d$ are the standard basis vectors. Similarly, we assume each $\boldsymbol{y}^{(i)}$ also lies in the standard probability $k$-simplex.

## 2.2 SIMPLICIAL APPROXIMATION

Simplicial approximation theorem allows the approximation of continuous functions using simplicial mappings. Before stating the theorem, we borrow a few definitions from the algebraic topology literature.

**Definition 1.** *(simplicial complex) A simplicial complex $K$ is a set of simplexes such that: 1) Every face of a simplex of $K$ is in $K$. 2) The intersection of any two simplexes in $K$ is a face of each of them. (A simplicial complex can be informally defined as a set of simplexes glued together through their faces.)*

**Definition 2.** *(simplicial mapping) A mapping between two simplicial complexes is called simplicial mapping if the images of the vertices are vertices.*

We also use the definition of a standard subdivisioning method which is used to break a $d$-simplex (or any other simplicial complex) into arbitrary small simplexes with the same dimension.

**Definition 3.** *(barycentric subdivision) Barycentric subdivision (BCS) of a $d$-simplex $K$ consists of $(d + 1)!$ $d$-simplexes. Each $d$-simplex $[\boldsymbol{v}_0, \boldsymbol{v}_1, ..., \boldsymbol{v}_d]$ out of these $(d + 1)!$ simplexes is associated with a permutation $\boldsymbol{p}_0, \boldsymbol{p}_1, ..., \boldsymbol{p}_d$ of the vertices of $K$ such that $\boldsymbol{v}_i$ denotes the barycenter (centroid) of $\boldsymbol{p}_0, \boldsymbol{p}_1, ..., \boldsymbol{p}_i$ where $1 \leq i \leq n$.*

Figure 2(a), (b) visualize examples of BCS for a 2-simplex, and a 3-simplex. Note that $i$-th BCS is the result of applying BCS to each simplex in the $(i - 1)$-th BCS. Using these definitions, simplicial approximation theorem can be stated as follows,

**Theorem 1.** *(**Simplicial Approximation Theorem**) Let $X$ and $Y$ be two simplicial complexes and $f : X \to Y$ be a continuous function. Then for an arbitrary $\epsilon$, there exist sufficiently large $k$ and $l$ and a simplicial mapping $g : X^{(k)} \to Y^{(l)}$ approximating $f$ such that $sup_{x \in X}\|f(x) - g(x)\| < \epsilon$. $X^{(k)}$ and $Y^{(l)}$ represent the $k$-th and $l$-th barycentric subdivision of $X$ and $Y$, respectively.*

In the appendix A, we provide a short topological proof for this theorem. Although the theorem provides a general approximation framework, its approximation is through using BCS of the input and output spaces. Each time BCS is applied, the number of simplexes is multiplied by $(d + 1)!$ and simplexes in higher order subdivisions become flatter and flatter Diaconis & Miclo (2011). This fact limits the use of this subdivision algorithm in practice. Moreover, BCS subdivides input or output space completely independent of the data. In the next section, we modify the BCS to a data-driven approach which allows us to learn a suitable subdivision given data.

Apart from BCS, in TDA, building simplicial complexes from data is often based on deterministic approaches. For instance, *Vietoris-Rips complex* Carlsson et al. (2006); Attali et al. (2013) is the set of all simplexes which their vertices are data points that their pair distances are less than a threshold. *Čech complex* Attali et al. (2013); Kerber & Sharathkumar (2013), is the set of all those simplexes that completely lie in a closed ball with specific radius. While these non-parametric data driven

methods can extract rich topological structures of the data using mathematical tools such as persistent homology, they are not often used as standard features in machine learning algorithms immediately Chazal & Michel (2017); Attali et al. (2013). In the next section, we modify BCS to a parametric framework that allows automation of the subdivisioning process and it can be used directly during the training of an SCN.

## 2.3 Automatic Subdivision

In the Algorithm 1, we have shown the process of generating a random simplex from the set of all simplexes in BCS of a $d$-simplex. The algorithm gives an uncommon view for identification of a simplex in the BCS which is through a set of nested simplexes. We modify this view to a data driven approach in a way that BCS is a special case of the modified version.

In the Algorithm 1, for a random permutation $P$, $N_0 \supset N_1 \supset ... \supset N_d$ form a nested set of simplexes. This fact is visualized in Figure 2(c), (d). $N_d$ indicates the simplex in the BCS that corresponds to $P$. Knowing the nested sequence uniquely determines a simplex in the BCS. Also, we note that each $N_i$ can be obtained by replacing one of the vertices of $N_{i-1}$ with a new vertex inside or on the boundary (closure) of it. Thereby, since the new vertex is in the closure of $N_{i-1}$, it can be represented as a convex combination of the vertices of $N_{i-1}$. The weights of these convex combinations can be computed in a straightforward way as shown in the Algorithm 1. Note that these weights are specified to the BCS.

To build a data driven approach initially we eliminate two restrictions of the Algorithm 1. First, we allow the number of repeats to be any arbitrary integer $l$ rather fixing it to $d$. This number is later referred as the subdivision depth. Second, at each repeat, we let the weights to freely have any arbitrary values as long as they can be used as a convex combination weights. This removes the restriction of having deterministic weights specified for BSC. This modification allows us to learn the weights through an optimization process.

A natural approach to make the subdivision process data-driven is to, instead of randomly sampling a simplex as in the Algorithm 1, sample a simplex with the probability proportion to the likelihood that it contains data and update the subdivision parameters accordingly. However, the number of all possible simplexes grows exponentially as $l$ increases, thereby making the computation of all these probabilities intractable. Alternatively, we sample from data and identify the simplex in the subdivision that contains the sample and use that simplex in an stochastic optimization framework to update the subdivision towards a desired one. But how can we identify the simplex that contains a given sample? Our parameterization of the subdivisioning process using the nested sequence of simplexes along with the following lemma help to locate the input using a forward pass,

**Lemma 1.** *Let $x$ denote a sample inside a $d$-simplex $\sigma = [v_0, ..., v_d]$. Assume $x = \sum_{i=0}^{d} w_i v_i$ is represented as a convex combination of the vertices of $\sigma$. Let $h = \sum_{i=0}^{d} w'_i v_i$ be another point in the simplex written in the same format. Indicating the simplex with vertices of $\sigma$ except $v_i$ replaced by $h$ with $\sigma^{-i}$, $x$ lies in the simplex $\sigma^{-j}$ where $j = \arg\min_i \frac{w_i}{w'_i}$.*

*Proof.* $x$ can be represented with the following convex combination which results the lemma,

$$x = (\sum_{i \neq j}(w_i - \frac{w_j}{w'_j}w'_i)v_i) + \frac{w_j}{w'_j}h$$

$\square$

Assume a sample $x = \sum_{i=0}^{d} w_i v_i$ represented as a convex combination of vertices of a given $d$-simplex $\sigma = [v_0, v_2, ..., v_d]$. Similar to the steps of Algorithm 1 and based on our view for identification of the target simplex through a nested set of simplexes, at the first step, a point is added inside the simplex using a convex combination of vertices of $\sigma$ with weights $w_1$. Then lemma 1 can be used to locate the $x$ in the $\sigma$ and accordingly replace one of its vertices with the new vertex. This process is repeated with the new $d$-simplex up to $l$ times to extract the simplex in the subdivision that contains $x$. Having the final simplex and a given cost function, parameters of the subdivision can be updated accordingly using the backpropagation and gradient descent framework. The procedure is formally shown in the Algorithm 2.

We refer the general procedure described in Algorithm 2 to as *automatic subdivision*. Note that the barycentric subdivision can be a particular outcome of the automatic subdivisioning.

---

**Algorithm 1** Generating a simplex in barycentric subdivision of a $d$-simplex

**input:** $d$-simplex $\sigma = [\boldsymbol{v}_0, \boldsymbol{v}_2, ..., \boldsymbol{v}_d]$, permutation $P = (p_0, p_1, ..., p_d)$

**initialize:** $N_0 = \sigma, \boldsymbol{w} = \mathbb{1}^d, j = 0$

**repeat**
    compute $\boldsymbol{u} = \sum_{i=0}^{d} \frac{w_i}{(d+1)-j} \boldsymbol{v}_i$
    Set $w_{p_j} = 0$
    Set $N_{j+1} = N_j$ with $p_j$-th vertex replaced by $\boldsymbol{u}$
    $j = j + 1$
**until** $j = d$

**return:** $N_d$

---

**Algorithm 2** One gradient step in automatic subdivisioning of a $d$-simplex using one data sample

**input:** $d$-simplex $\sigma = [\boldsymbol{v}_0, \boldsymbol{v}_2, ..., \boldsymbol{v}_d]$, sample $\boldsymbol{x} = \sum_{i=0}^{d} w_{x_i} \boldsymbol{v}_i, \Theta = \{\boldsymbol{w}_1, ..., \boldsymbol{w}_l\}$, Loss $L$
**initialize:** $N_0 = \sigma, j = 0$
**repeat**
    Compute $\boldsymbol{u} = \sum_{i=0}^{d} w_{j_i} \boldsymbol{N}_{j_i}$
    Set $k = arg \min_i \frac{w_{x_i}}{w_{j_i}}$
    Set $N_{j+1} = N_j$ with $k$-th vertex replaced with $\boldsymbol{u}$
    Set $\boldsymbol{w_x}$ as convex combination weights of $x$ represented with vertices in $N_{j+1}$
    $j = j + 1$
**until** $j = l$
$\Theta = \Theta - \alpha \nabla_\Theta L(\Theta)$
Project each $\boldsymbol{w_i} \in \Theta$ on the standard $d$-simplex

---

## 2.4 SIMPLICIAL COMPLEX NETWORKS

As theorem 2 states, any continuous function from a simplicial complex to another can be approximated with a simplicial mapping from BCSs of the input space to BCSs of the output space. In the last section we explained how we can subdivision the input space through the automatic subdivision process where BCS was a particular outcome. In this section, we develop SCNs to parameterize a class of simplicial mappings with trainable values on vertices of the input space subdivision and it is defined linearly on the $d$-simplexes of the subdivision using the evaluations at its vertices, resulting a piece-wise linear simplicial mapping. Parameters of this simplicial mapping are then optimized for function approximation or a regression tasks. We leverage a same technique used in the previous section to parameterize the SCN output.

Recalling our initial assumption that inputs lie in a $d$-simplex $\sigma = [\boldsymbol{v}_0, ..., \boldsymbol{v}_d]$ ($\boldsymbol{v}_0$ representing the origin and other $\boldsymbol{v}_i$ are the standard basis vectors), a given input $\boldsymbol{x}$ can be represented with the following convex combination,

$$\boldsymbol{x} = (1 - \sum_{i=1}^{d} x_i)\boldsymbol{v}_0 + \sum_{i=1}^{d} x_i \boldsymbol{v}_i \tag{1}$$

In the previous section, we showed that how the position of $\boldsymbol{x}$ can be found in $\sigma$ through the subdivisioning process. SCN's output at $\boldsymbol{x}$ is calculated using the values of its mapping at the vertices of $\sigma$ and added vertices during locating $\boldsymbol{x}$. Simplicial mapping at the $m$-th added vertex is defined recursively using its preceding vertices. Assuming that the $m$-th added vertex represented as $\boldsymbol{h_m} = \sum_{i=0}^{d} w_{m,i} \boldsymbol{u}_i$, where $\boldsymbol{u}_i$ are vertices of the preceding simplex in the nested sequence of simplexes used to locate $\boldsymbol{x}$ and $w_{m,i}$ are its corresponding convex combination weights, the value of the simplicial mapping at $\boldsymbol{h_m}$ is defined as,

$$f(\boldsymbol{h_m}) = \sum_{i=0}^{d} w_{m,i} f(\boldsymbol{u}_i) + b_m(\boldsymbol{h_m}) \tag{2}$$

In other words, $f(\boldsymbol{h_m})$ is defined using a convex combination of $f(\boldsymbol{u}_i)$ with the same weights, added with a bias which is a function of $\boldsymbol{h_m}$. A geometrical view of equation 2 for a 2-simplex is shown in the Appendix C. Using our proof of simplicial approximation theorem in the appendix A, it is straightforward to show that as long as we consider each $b_m$ is an arbitrary function of the $\boldsymbol{h_m}$, the simplicial mapping holds the universal approximation property. In our experiments, however, we

used a primitive model for biases and defined them as constant parameters. We will empirically show that even this simple model for biases can result complicated simplicial mappings and accurate approximations.

All in all, an SCN is defined using the mentioned process (Figure 1 visualizes a general architecture). Parameters that should be learned are bias function parameters, and the weights that we used to subdivision the input space. All these parameters can be updated using the backpropagation algorithm. Another point that must be noted here is that, as shown in Figure 1, inputs to $\boldsymbol{h}_2, ..., \boldsymbol{h}_l$ are not specified. Even though we described how to obtain the value of these inputs using lemma 1, the order that the vertices of the preceding simplex are combined is not specified. We refer to the policy on the ordering of the vertices used to fed to the next convex combination as the SCN's *network policy*.

Specifying the depth, network policy, and the bias functions fully determines the architecture of an SCN. To train an SCN, the derivative of the loss function over mini-batches is taken with respect to the weights and the bias function parameters, and these parameters are updated using a gradient descent approach such as stochastic gradient descent, or Adam Kingma & Ba (2014). General training process for an SCN is shown in the Algorithm 3. Note that after updating weights of the network, for each hidden node, a projection of weights is used such that their summation is equal to 1. This projection can be avoided through use of logits and the Softmax function in parameterizing the weights.

---

**Algorithm 3** training procedure for a general SCN

---

$S = [\boldsymbol{v}_0, ..., \boldsymbol{v}_d]$, $l$ = depth, $\theta_b, \theta_W$ (bias function, and weight params), $P$ (network policy), $(\boldsymbol{x} = \sum_{i=0}^{d} w_{x_i} S_i, y)$ (input/output pair), $\alpha$ (learning rate)
*// forward pass*
**for** $m \in \{1, ..., l\}$ **do**
   Permute $S$ using $P$
   $\boldsymbol{h}_m = \sum_{i=0}^{d} w_{m,i}.S_i$
   $f(\boldsymbol{h_m}) = \sum_{i=0}^{d} w_{m,i} f(S_i) + b_m(S; \theta_b)$
   Extract $j$ and update $\boldsymbol{w}_x$ using $\boldsymbol{x}, S$, and lemma 1
   $S_j = \boldsymbol{h_m}$
**end for**
$f(\boldsymbol{x}) = \sum_{i=0}^{d} w_{x_i} f(S_i)$
*// backward pass and parameter updates*
$\theta_b = \theta_b - \alpha \nabla_{\theta_b} \mathcal{L}(f(\boldsymbol{x}), y)$
$\theta_W = \theta_W - \alpha \nabla_{\theta_W} \mathcal{L}(f(\boldsymbol{x}), y)$
**for** $m \in \{1, ..., l\}$ **do**
   project $\boldsymbol{w}_m$ on the standard $d$-simplex
**end for**

---

## 3 RELATED WORK

In TDA, *Persistent homology* methods are commonly used to extract features that are robust to perturbations of the input Otter et al. (2017); Adams et al. (2017). A range of works use these features in a feed-forward architecture. For instance, in Liu et al. (2016), *persistence landscape*, a topological summary of data, is used to develop persistent layers. These layers are used in a neural network architecture along with convolutional layers and the resulting architecture is applied to the problem of music tagging Law et al. (2009), and multi-label classification Firouzi et al. (2017). A similar approach is applied in Umeda (2017) for the time series classification Xi et al. (2006). Cang & Wei (2017) introduces TopologyNet, an architecture that uses a persistent homology method for 3D biomolecular structures to extract features suitable for convolutional layers. Hofer et al. (2017) proposes a trainable layer to learn suitable representation from *persistence diagram* of the data. This layer which extracts topological signatures from the data is used along with a common neural network architecture and their approach achieved the state-of-the-art in a specific task on classification of social network graphs at the time.

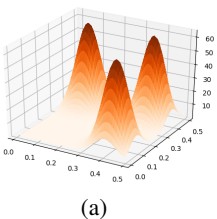 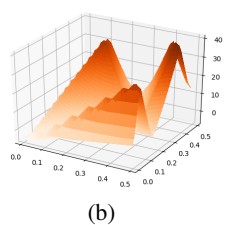 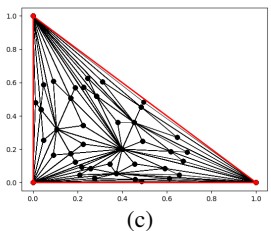

(a)          (b)          (c)

Figure 3: Approximation of summation of two dimensional Gaussian functions with an SCN. (a) target function, (b) approximated function using an SCN with a depth of 4, (c) the learned subdivision on the input space.

Although aforementioned methods try to improve the performance of neural networks through using topological scenarios, the TDA geometrical perspective vanishes as they are aligned with commonly used architectures. In addition, a specific persistent homology method is applied for a determined task. SCNs are single-model architectures that can be trained with the common forward and backward passes. In addition, no specific persistent homology is used to extract the topological geometry of the inputs, which enables its generalization to other domains.

## 4 EXPERIMENTS

We perform regression tasks in order to evaluate the performance and the complexity of functions that SCN can learn. Mean squared error was used as the loss function in all the experiments. In some cases, an SCN with a few number of parameters can estimate a function better than a neural network with fully connected layers even with around 10 times more parameters. In the appendix B, with a straightforward derivation, we also show that how does an SCN without a hidden unit reformulates the linear regression problem.

### 4.1 APPROXIMATING SUM OF GAUSSIAN FUNCTIONS

As a primary experiment, we approximate sum of three Gaussian functions with an SCN with a depth of 4 and constant parameters as the biases. The resulting simplicial mapping and the learned subdivision is shown in Figure 3. [1]

### 4.2 TWO MORE TOY EXPERIMENTS

We compare the performance of using an SCN in the problem of learning particular structures to the results obtained by a neural network with fully-connected layers and ReLU activations. This is a more general experiment done in Anil et al. (2018) to learn the absolute value function. The experiment was used in Anil et al. (2018) to show the limitations of specific activation functions for neural networks in learning some of the basic functions. Here, we show that SCN can learn more complicated structures even with constant parameters as its bias functions. Models are trained using Adam Kingma & Ba (2014) with learning rate 0.001, and a same mini-batch size. Figure 4 represents the comparison. More details about the experiments, and a binary classification experiment on the MNIST dataset can be found in the appendix F.

### 4.3 MEMORY ANALYSIS

Several works have proposed approaches to improve the memory efficiency of training deep and very deep neural networks Behrmann et al. (2018a); Srivastava et al. (2015); Chen et al. (2016); Gomez et al. (2017). Particularly, a reversible architecture Chang et al. (2018); Behrmann et al. (2018a); Gomez et al. (2017) may result an $O(1)$ memory cost of activation in terms of the number of layers. Although due to non-existence of activation function in SCNs the term reversibility is not applicable

---

[1]More accurate approximations and complicated subdivisions using deeper SCNs is provided in the ANONY-MOUS link.

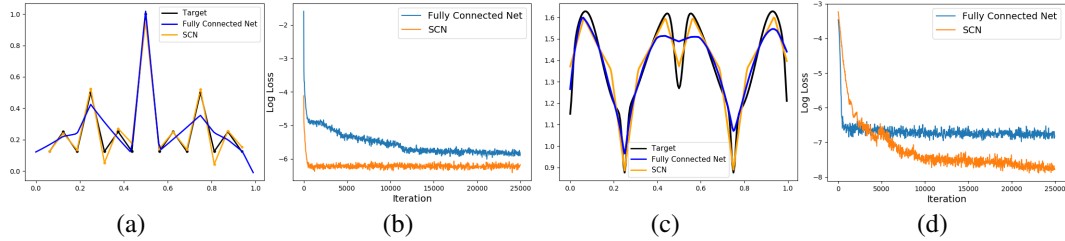

Figure 4: Comparison of an SCN with a fully-connected neural network architecture in approximation of specific functions: (a) and (c) indicate the target and fitted functions, (b) and (d) indicate the comparison in terms of log loss.

here, to backpropagate through the SCN's architecture it is required to store only $d$ out of $d + 1$ vectors and their function evaluations, which were used to extract the last hidden node and evaluate its function value, latest convex weights for the input in the forward pass, and an array of integers with a length of $l$ indicating the order that nodes were removed during the forward pass. Storing these values is enough for the gradient computation of the network parameters. Interestingly, it is not needed to store the weights of the network and function evaluations at the previous nodes as they all can be extracted with these information (proof in the Appendix E). This approach trades the cost of memory with an additional computational cost and can be helpful in training very deep SCNs.

## 5 LIMITATIONS

In some conditions, the bias function for a hidden node may outputs high norm vectors. In these conditions, output may not be close to a smooth curve. Accordingly, SCN's output not behave well in practice and potentially overfits to data. In case of using a conventional neural networks, to increase the stability of the output, one approach is to enforce Lipschitz constraints Anil et al. (2018); Arjovsky et al. (2017); Gouk et al. (2018); Behrmann et al. (2018b). Enforcing Lipschitz constraints in an SCN can be done through their bias functions. The simplest approach is perhaps to clip the values of the bias functions to a predetermined interval.

In some cases, input weights to a hidden node might converge to degenerate values with the value one in one position and zero elsewhere. In these cases, the corresponding layer does not change the learned subdivision and it may be assumed as a redundant layer. We refer to this phenomena as *weight drifting*. In situations that we have a deep SCN architecture with higher than the true required capacity, these layers with a bias value close to zero may be helpful to automatically adjust the capacity of the model and prevent SCN from overfitting.

Throughout this work we assumed that inputs lie in a given $d$-simplex. The assumption was used just for the sake of presentation. It can be assumed that samples lie in any given simplicial complex, not a specific $d$-simplex. The vertices of any simplex in the simplicial complex that a sample falls in can be used as the primary nodes in the network.

Choosing an appropriate network policy for SCNs may be assumed as a challenging task in different domains. In our experiments, we observed that even simple policies result a proper function approximation. In fact, in one dimensional case, a direct use of the Stone-Weierstrass theorem can be used to prove that SCNs are universal approximators even with random policies and fixed weights. In our experiments, the sensitivity of SCNs to the choice of their bias function observed to be larger than the network policy.

## 6 CONCLUSION AND FUTURE WORK

In this work, we have used techniques from topological data analysis to build a class of neural network architectures with the universal approximation property which can be trained using the common neural network training framework. Topological data analysis methods are based on the geometrical structure of the data and have strong theoretical analysis. SCNs are made using the geometrical view

of TDA and we believe that they can be used as a step towards building interpretable deep learning models.

Most of the experiments in the paper are synthetic. More practical applications of the paper is considered as an immediate continual work. Moreover, throughout this work, bias functions of the simplest kinds (constant parameters) were used. We mentioned earlier that a bias function may be an arbitrary function of its input to keep the universal approximation property of SCNs. A natural idea is to use common neural network architectures as the bias function. In this case, backpropagation can be continued to the bias function parameters as well. This is also considered as another continuation of this work.

ACKNOWLEDGMENTS

Anonymous

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

## A    PROOF OF SIMPLICIAL APPROXIMATION THEOREM

We provide a short topological proof of the approximation theorem we used in main text. Some mathematical precision is dropped throughout the proof.

**Theorem 2.** *(Simplicial Approximation Theorem) Let $X$ and $Y$ be two simplicial complexes. Let $f : |X| \to |Y|$ be a continuous function. Then for an arbitrary $\epsilon$, there exist sufficiently large $k$ and $l$ and a simplicial mapping $g : |X^{(k)}| \to |Y^{(l)}|$ approximating $f$ such that $sup_{x \in X} \|f(x) - g(x)\| < \epsilon$.*

*Proof:* In order to prove this theorem we use the following definition:

**Definition 4.** *(Star) Let $K$ be a simplicial complex and $v \in K$ a vertex of $K$. The star of $v$ represented by $St(v)$ is defined as the union of interior of all simplexes of $K$ that contain $v$ as a vertex. Note that $St(v)$ is an open set. Figure 5 pictures an example for star of a vertex.*

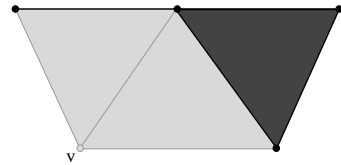

Figure 5: Star of a vertex $v$ of a simplicial complex (gray region)

Choose $l$ in order that the diameter of simplexes of $Y^{(l)}$ is less than $\epsilon$. Let $W$ denotes the $Y^{(l)}$ vertices. Since $f$ is a continuous function, pre-image of an open set in an open set. Therefore, for each $w \in W$, $f^{-1}(St(w))$ is an open set. $\{f^{-1}(St(w))\}_{w \in W}$ is an open covering for $|X|$. Let $\delta$ represents the Lebesgue number of this covering. Choose $k$ such that the diameter of simplexes of $X^{(k)}$ be less than $\frac{\delta}{2}$. Denote the vertices of $X$ with $V$. It can be shown that for each $v \in V$, diameter of $St(v)$ is less than $\delta$ (proof is shown in figure 6). So, for each $v \in V$, there exist a $w \in W$ such that $St(v) \subset f^{-1}(St(w))$. We define the simplicial map $g$ such that $g(v) = w$. Also, note that $f(St(v)) \subset St(w)$.

Now we prove that $g$ approximates $f$ as desired. Let $\sigma \in X^{(k)}$ be a simplex in $A$. Let $v_1, v_2, ..., v_p$ denote the vertices of this simplex. For any $x \in \sigma$ and a vertex $v_i, i \in \{1, 2, ..., p\}$, $x$ is in the $St(v_i)$. So, $f(x)$ is in the $\cap_{i=1}^{p} f(St(v_i))$. Using the last note in the previous paragraph, we have $x \in \cap_{i=1}^{p} St(g(v_i))$. This fact means that $f(x)$ lies in the simplex that its vertices are $f(v_1), f(v_2), ..., f(v_p)$.

We now extend the definition of $g$ for all non-vertex elements of $X$. Let $x \in |X|$ be a non-vertex element represented as a convex combination of a number of vertices of $V$ as $x = \Sigma t_i v_i$. We define $g(x)$ as,

$$g(x) = g(\Sigma t_i v_i) = \Sigma t_i g(v_i)$$

Straightforward steps can be used to prove that $g$ is a continuous simplicial mapping. Using the facts in the last two paragraphs, we conclude that for each $x \in |X|$ and a simplex $\sigma \in X$ containing $x$, both $f(x)$, and $g(x)$, lie in the simplex with vertices $\{g(v)\}_{v \in \sigma}$. Due to the initial choice of $l$, we know that diameter of this simplex is less than $\epsilon$, which means $sup_{x \in X} \|f(x) - g(x)\| < \epsilon$.    □

## B    REFORMULATION OF LINEAR REGRESSION PROBLEM USING AN SCN WITHOUT HIDDEN NODES

In case that an SCN has no hidden node (no subdivisioning process), it can be viewed as linear regression reformulation. A real valued linear function $f : \Delta^d \to R$ from a $d$-dimensional simplex $\Delta^d = [v_0, ..., v_d]$, can be specified by the values of $f$ at each $v_i$. These values are represented by $f(v_i)$.

Assume a data matrix $X \in \mathbb{R}^{N \times d}$ of $N$ samples within $\Delta^d$, and their corresponding output in a vector $y$. We formulate the linear regression problem with training a weight $w$ that minimizes $||Xw - y||_2^2$.

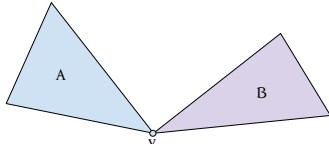

Figure 6: Let $A, B$ be two simplexes containing $v$. Using triangle inequality for a distance function $d$, for each $a \in A, b \in B$ we have: $d(a, b) \leq d(a, v) + d(v, b) < \frac{\delta}{2} + \frac{\delta}{2} < \delta$. This fact shows that the diameter of $St(v)$ is less than or equal to $\delta$.

We represent the coefficients of representing samples in $X$ as a convex combination of $\boldsymbol{v}_0, ..., \boldsymbol{v}_d$ in a matrix $C \in \mathbb{R}^{N \times (d+1)}$ with a rank of at most $d$, where $i$-th row indicates the corresponding coefficients for $i$-th sample. Then the linear regression problem can be reformulated as,

$$\|C\boldsymbol{f} - \boldsymbol{y}\|_2^2$$

where $\boldsymbol{f}$ is a $(d+1)$ dimensional vector representing the function value at $v_i$ as its $i$-th element. With a straightforward computation, One can verifies that the optimal $\boldsymbol{w}$ or $\boldsymbol{f}$ can be computed from the optimal value of the other one.

## C  GEOMETRIC VISUALIZATION OF EQUATION 2

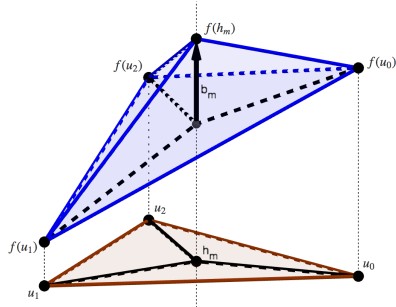

Figure 7: A geometrical view of how does equation 2 evaluates the simplicial mapping at $m$-th added point using its preceding vertices $\boldsymbol{u}_i$. A same convex combination of $\boldsymbol{u}_i$ used to generate $\boldsymbol{h}_m$ is applied to the corresponding $f(\boldsymbol{u}_i)$. This value added with a bias term determines $f(\boldsymbol{h}_m)$.

## D  PROOF OF LEMMA 1

$x$ can be represented with the following convex combination which results the lemma,

$$\boldsymbol{x} = \left(\sum_{i \neq j}(w_i - \frac{w_j}{w'_j}w'_i)\boldsymbol{v}_i\right) + \frac{w_j}{w'_j}\boldsymbol{h}$$

## E  RETRIEVING PREVIOUS LAYERS PARAMETERS

In contrast to Dinh et al. (2014); Gomez et al. (2017), no separation of input into blocks is needed for SCNS to extract outputs of the previous layers. Retrieving the values for previous in SCN is a result of the fact that knowing all weights, all vertices except one, and the resulting vector of a convex combination is enough to extract the missing vertex. Formally, let $\boldsymbol{x} = \sum_{i=0}^{d} w_i \boldsymbol{v_i}$ represents a convex combination ($\sum_{i=0}^{d} w_i = 1$, and $\forall 0 \leq i \leq d : w_i \neq 0$). Assuming $\boldsymbol{v_j}$ is unknown, it can be

computed simply as,

$$v_j = \frac{1}{w_j}(x - \sum_{i \neq j} w_i v_i)$$

.

Assume an SCN with a depth of $l$. Given the SCN's last layer weights $w_l$, the bias functions $b_1, ..., b_l$, and its network policy $P$, the following algorithm shows that knowing the last $d$ hidden nodes, their function values, and an array indicating the order of vertices that was removed during the forward pass to locate $x$, is enough to extract all the previous function values, hidden nodes, and also the weights. For simplicity of the algorithm, $u_0, ..., u_{l+d}$ is used as indicators for $v_0, ..., v_d, h_1, ..., h_l$ with the same order.

---

**Algorithm 4** Retrieving preceding layers and parameters for the backward pass of an SCN

---

given $u_{l+d}, u_{l+d-1}, ..., u_{l+1}$, $f(u_{l+d}), f(u_{l+d-1}), ..., f(u_{l+1})$, and convex weights $w_x$ of representing the input $x$ as the convex combination of $u_{l+d}, u_{l+d-1}, ..., u_l$, array $a = [a_1, ..., a_{l-d}]$ of integers storing the indices of $u_i$ were removed during the forward pass with the exact order.

$j = 0$

**repeat**

$u_{l-j} = \frac{1}{w_x^{(u_{l-j})}}(x - \sum_{i=0}^{d-1} w^{(u_{l+d-i})} u_{l+d-i})$

Extract $w_{l-j}$ as the convex combination weights of representing $u_{l+d-j}$ using $u_{l+d-j-1}, ..., u_{l-j}$

Extract $w_{l-j}^{(u_{l+d-j-i})}, (\forall 0 \leq i \leq d)$ (element of $w_{l-j}$ that is assigned to $u_{l+d-j-i}$) using the network policy $P$, and $a_{l-d-j}$.

$b_{temp} = b_{l-j}(h_{l-j}, ..., h_{l+d-j})$

$f(h_{l-j}) = \frac{1}{w_{l-j}^{(u_{l+d-j})}}(f(x) - b_{temp} - \sum_{i=1}^{d} w_{l-j}^{(h_{l+d-j-i})} f(h_{l-j-i}))$

Update $w_x$ to the convex combination weights of $x$ represented using $u_{l+d-j-1}, ..., u_{l-j}, u_{l-j-1}$

$j + = 1$

**until** $j = d - 1$

---

## F    EXPERIMENTAL DETAILS

We provide the details of the experiments in the main text as well as results of a binary classification experiment on MNIST as a primary proof of concept for practical usages of SCNs.

In all the three toy experiments, the neural network consisted of two fully connected layers with 300 and 2 hidden neurons respectively. ReLU activations was used for both layers. The SCN model had a depth of 4. Accordingly, the network had 4 bias functions which all were constant parameters. A learning rate of 0.001, mini-batches of size 100 were used. The network policy was random. Thereby, preceding nodes were assigned randomly to the next convex combination weights (recall the fact that in one dimensional case, SCNs are universal approximators even with a random policy and fix convex combination weights. Thereby the optimization can be done through their biases only). In the sum of Gaussians experiment, inputs lied in a 2-simplex with vertices $(0, 0), (0, 1)$, and $(1, 0)$ as the way that SCNs are presented throughout the paper. Similarly, in the one dimensional case inputs lied in the interval (1-simplex) $[0, 1]$.

To provide an observation that SCNs can be used in practice, we did a primary experiment on the MNIST data set. The aim was the binary classification of zeros and ones. We lessened the dimensionality of the data to 20 using PCA. We compared SCN to a simple logistic regression, and a neural network with the same architecture used in our synthetic experiments. Logistic regression can be seen as an SCN without any hidden nodes followed by a Sigmoid. We used a single unit SCN where its bias function was again a single constant parameter. Models were trained using the binary cross entropy loss and a same learning rate and mini-batch size were used in training the models. The

performance is shown in the Figure 8 and the test accuracy and the number of parameters are shown in the Table 1. Although the performance of the SCN with a single hidden unit is not as good as the neural network with fully connected layers, SCN could improve the accuracy of logistic regression by around 7% with adding a single hidden node to its architecture. Scaling up the SCN architecture to higher dimensional spaces is considered as a continuation of the paper.

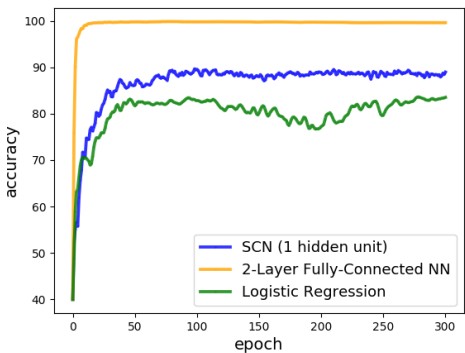

Figure 8: Accuracy of the models on the binary classification of zeros and ones in MNIST.

Table 1: Binary classification on MNIST

| Model | # of parameters | accuracy |
|---|---|---|
| 2-layer Fully Connected NN | 6300 | $99.61 \pm 0.16$ |
| SCN (1 hidden unit) | 41 | $89.00 \pm 1.01$ |
| Logistic Regression | 21 | $82.11 \pm 2.85$ |

