# OpenReview forum: "Simplicial Complex Networks"
_ICLR.cc/2020/Conference — Reject_

### Official Review · AnonReviewer1 · 2019-10-20
**Official Blind Review #1**

**Rating:** 1

**Review:**

1. Summary of the paper

This paper introduces *simplicial complex networks*, a new class of
neural networks based on the idea of the subdivision of a simplicial
complex. Using the simplicial approximation, a classical theorem in
algebraic topology, the paper demonstrates that the network is capable
of learning a suitable function approximation that in turn can be used
for regression tasks.

2. Summary of the review

This paper brings an interesting perspective to the table, viz. the use
of concepts from algebraic topology to develop new neural network
architectures with improved approximation capabilities. I am
appreciative and excited of any work in this direction, yet I deem the
current version of this paper not acceptable for publication because of
the following issues:

- The paper suffers from a lack of clarity that makes assessing the
  results and the proposed architecture harder than it needs to be.
  In particular, numerous aspects such as scalability or the
  generalization capabilities are glossed over. Even though I am
  familiar with the underlying concepts and techniques, I found the
  exposition not always easy to follow.

- The experiments in the paper appear to be highly preliminary and need
  more refinement. I understand that it is tough to develop a new
  technique and 'compete'  against so many already existing ones, but
  nevertheless I would suggest to at least extend one of the experiments
  to a larger-scale comparison. Personally, I found the function
  approximation experiment to be very interesting, so to me, this would
  be a natural starting point for a more in-depth comparison.

Again, I want to stress that I think this paper has the potential to
become a strong contribution but for this to happen, it requires an
additional revision cycle. I will outline some suggestions below.

3. Clarity & technical correctness

It is my understanding that the paper is technically correct on all
levels. At times, imprecise language is used that might mislead or
confuse readers.

My overall recommendation would be to start the paper with a brief
intuition of _why_ BCS is required and how to make use of the
approximation theorem. After this, I would propose a high-level overview
of the new architecture and its individual components. Only _then_ would
I suggest describing each of these components in more detail.

Here are some detailed suggestions:

- The flow of the abstract could be improved; it is not clear why TDA
  methods are suitable here.

- Topological data analysis is not directly a 'geometric approach' for
  data analysis. It incorporates specific attributes of geometry by
  means of filtrations, for example, to 'escape' the coarseness of
  unweighted simplicial complexes

- TDA is also not _just_ restricted to point clouds

- I find the difference between 'deterministic' and 'statistical'
  methods incorrect in the introduction; the kernel-based approaches
  that were developed to compare persistence diagrams, for example, are
  _not_ the opposite of 'deterministic'. I would rather make
  a distinction between 'distance-based methods' that are hard to
  compute, and 'kernel-based methods' with improved scalability.

- When analysing point clouds, TDA does not use 'an inducing' distance;
  it can make use of the induced distance of a graph, for example, but
  typically, the distances are pre-defined and evaluated between all
  pairs of points.

- I find the statement that hidden units are conceptually well-defined
  only in SCNs to be highly problematic; I fully agree that
  interpretability of architectures is relevant, but another expression
  should be used here---FCN architectures (or in fact, all other
  architectures in deep learning) are perfectly well-defined on the
  conceptual level. Maybe the paper should be more clear about what it
  expects the 'role' of a hidden unit to be; in fact, this is also not
  discussed satisfactorily at present.

- Introduce the concept of a *simplex* briefly before discussing
  simplicial complexes.

- Figure 1 cannot be understood in the context in which it is placed
  currently; either move it back or create another figure that is more
  high-level and explains the concepts

- The requirement that all data points are within the standard
  probability simplex seems vacuous to me; this can always be achieved
  by scaling. Am I misunderstanding this? I would rewrite this and make
  it clear that this is _not_ a limiting assumption.

- Definition 2 is lacking formality; please add a brief description of
  the function itself

- Definition 3 is could use some more intuition; I understand that it
  is technically correct, but a brief illustration of the iterative
  nature of subdivision would be helpful.

- After introducing the simplicial approximation theorem, please give
  a brief connection on its relevance to the problem at hand.

- The brief introduction of the Vietoris--Rips complex is relatively
  hard to understand; please rephrase it. It is the complex in which
  a set of points is a simplex if and only if the pairwise distances
  of the points are less than or equal to the pre-defined threshold.

- Section 2.3 is highly confusing at first because it requires readers
  to understand Algorithm 1; please rephrase the introduction to this
  section to make it clear what the _purpose_ is. To my understanding,
  you want to use the BCS in order to obtain a situation as described
  by the approximation theorem---but at this point, the utility of the
  subdivision is not clear yet.

- What does it mean that weights are 'specified to the BCS'?

- When discussing the 'data-driven' approach, please add a motivation
  for this. I understand that such a subdivision is required in order
  to be computationally feasible, but this needs to be motivated.

- Lemma 1 appears to be a direct consequence of the definition of
  simplicial subdivision. I would suggest moving its proof to the
  appendix.

- Please give an improved explanation and intuition of the parameters of
  the subdivision and why they require updating. It took me a while to
  understand the relevance of the weights and what is actually updated
  in the algorithm.

- When talking about SCNs in Section 2.4, the salient points (learning
  a subdivision with trainable vertex function values etc.) need to be
  mentioned well before this section. When reading the paper for the
  first time, I started understanding the method better while reading
  the section, so I think that this intuition should become as early as
  possible.

- Please add intuition why _finding_ the position of a point is
  required; I understand that this is necessary in order to obtain
  weights and neighbours, from which to obtain a mapping, but at first,
  this 'location problem' struck me as artificial.

- When talking about the 'network policy', more details are needed. How
  relevant is this choice for the network?

- When discussing related work, please look at the follow-up
  publication from Hofer et al., which discuss more architectures and
  use cases:

      Hofer et al., Connectivity-Optimized Representation Learning via Persistent Homology
      ICML 2019

- 'Topological geometry' on p. 7 is an unfamiliar term to me. What is
  meant by this?

- What about the computational complexity? This is somewhat glossed over
  in the description of the BCS algorithm. Is this currently
  a bottleneck for the deploying the architecture?

- How easy would it be to extend this to _classification_ problems? The
  title suggests that the architecture is highly generic, yet the paper
  only discusses regression tasks. At the very least, this should be
  briefly discussed.

4. Experiments

- The experiments are lacking a brief explanation: what are they
  supposed to show or analyse? I find the function approximation
  example to be very interesting, but somewhat preliminary insofar
  they only compare *two* functions. Why not generate sequences or
  families of functions and make this into a large-scale comparison?

- I found the discussion of storage to be somewhat surprising; is this
  a property that the paper wants to stress for SCNs? If so, more
  experiments in that direction are required.

- The observations on bias function choice are not reflected in the
  experiments that are presented in the main text. I understand that it
  works for the toy examples, but a more in-depth comparison is required
  for this sort of experiment to be compelling.

- Is it possible to _tune_ the number of subdivisions or provide an
  upper bound? It would be interesting to see how such a bound might
  change the results---this is just a suggestion, but I feel that it
  might be interesting to discuss how much performance or approximation
  qualities are gained by _not_ restricting the BCS algorithm.

5. Minor style comments

The presentation quality of the paper could be improved. Here are some
suggestions.

- Please use `\citep` when citing works in the text; currently, citations are intermingled with sentences and make it harder to read the paper.
- 'Universal approximation property' --> 'The universal approximation property'
- 'with similar property' --> 'with similar properties'
- 'alternates the common architecture' --> 'changes the common architecture'
- the citation 'Bernardo et al.' should be fixed; it is missing information
- I suggest removing the sentences 'The rest of this paper is organized as follows'...
- 'Let f denotes a mapping' --> 'Let f denote a mapping'
- 'As theorem 2' --> 'As theorem 1'
- 'we can subdivision' --> 'we can subdivide'
- 'a same' --> 'the same'
- 'that how the position' --> 'how the position'
- 'used to fed' --> 'used to feed'
- 'the appendix F' --> 'appendix F'
- 'may outputs' --> 'may output'
- 'this phenomena' --> 'this phenomenon'
- 'bias function observed to be' -->' bias function was observed to be'
' 'can be continued' --> 'can be extended'

**Experience Assessment:**

I have published in this field for several years.

**Review Assessment: Checking Correctness Of Derivations And Theory:**

I carefully checked the derivations and theory.

**Review Assessment: Checking Correctness Of Experiments:**

I carefully checked the experiments.

**Review Assessment: Thoroughness In Paper Reading:**

I read the paper thoroughly.

---

### Official Review · AnonReviewer3 · 2019-10-22
**Official Blind Review #3**

**Rating:** 1

**Review:**

This paper presents a new neural network architecture called simplicial complex networks. The authors draw inspiration from topological data analysis in order to motivate this new architecture. The SCN architecture is motivated through a parametric sub-divisioning of the input space into a complex of d-dimensional simplices. The authors give a universal approximation theorem for a model of this type, and give a simple algorithm to perform the sub-divisioning. The resulting network output is a convex combination of vertices after l nested sub-divisions. The authors present a number of toy experiments to validate their claims.

Although I believe the authors present an intriguing idea, I ultimately tend for a rejection of this paper. I have three major reasons for this:
(1) Quality of writing and clarity is lacking: Citations are all done in-line, symbols are being used without proper explanation, and many sentences are just purely broken and impossible to understand.
(2) The authors do not properly compare their approach to the most common architectures in use: Fully-connected nets and CNNs. What are the shortcomings and benefits of SCNs as compared to these architectures?
(3) Usefulness of the approach. I believe this ties in with point (2). The authors make no explicit mention of the computational complexity of their approach, or how the number of parameters depends on the number of input dimensions. Since the simplicial complex is defined on the input space the number of parameters scales with the number of input dimensions, and becomes restrictively high. This explains why the author use only low-dimensional toy examples.

**Experience Assessment:**

I do not know much about this area.

**Review Assessment: Checking Correctness Of Derivations And Theory:**

I assessed the sensibility of the derivations and theory.

**Review Assessment: Checking Correctness Of Experiments:**

I carefully checked the experiments.

**Review Assessment: Thoroughness In Paper Reading:**

I read the paper at least twice and used my best judgement in assessing the paper.

---

### Decision · Program_Chairs · 2019-12-19

**Decision:**

Reject

**Comment:**

The aper introduces simplicial complex networks, a new class of
neural networks based on the idea of the subdivision of a simplicial
complex. The paper is interesting and brings ideas of algebraic topology to inform the design of new neural network architectures.

Reviewer 1 was positive about the ideas of this paper, but had several concerns about clarity, scalablity and the sense that the paper might still be in an early phase. Reviewer 2 had similar concerns about clarity, comparisons, and usefulness. Although there were no responses form the author, the discussion explored the paper further, but continued to think the idea is still in its early phase.

The paper is not currently ready for acceptance, and we hope the authors will find useful feedback for their ongoing reasearch.